# From Local Energy Communities towards National Energy System: A Grid-Aware Techno-Economic Analysis

Cédric Terrier *, Joseph René Hubert Loustau, Dorsan Lepour and François Maréchal

Department of Industrial Processes and Energy Systems Engineering, Institute of Mechanical Engineering, École Polytechnique Fédérale de Lausanne, Rue de l'Industrie 17, 1950 Sion, Switzerland; joseph.loustau@epfl.ch (J.R.H.L.); dorsan.lepour@epfl.ch (D.L.); francois.marechal@epfl.ch (F.M.)
* Correspondence: cedric.terrier@epfl.ch

**Abstract:** Energy communities are key actors in the energy transition since they optimally interconnect renewable energy capacities with the consumers. Despite versatile objectives, they usually aim at improving the self-consumption of renewable electricity within low-voltage grids to maximize revenues. In addition, energy communities are an excellent opportunity to supply renewable electricity to regional and national energy systems. However, effective price signals have to be designed to coordinate the needs of the energy infrastructure with the interests of these local stakeholders. The aim of this paper is to demonstrate the integration of energy communities at the national level with a bottom–up approach. District energy systems with a building scale resolution are modeled in a mixed-integer linear programming problem. The Dantzig–Wolfe decomposition is applied to reduce the computational time. The methodology lies within the framework of a renewable energy hub, characterized by a high share of photovoltaic capacities. Both investments into equipment and its operation are considered. The model is applied on a set of five typical districts and weather locations representative of the Swiss building stock. The extrapolation to the national scale reveals a heterogeneous photovoltaic potential throughout the country. Present electricity tariffs promote a maximal investment into photovoltaic panels in every region, reaching an installed capacity of 67.2 GW and generating 80 TWh per year. Placed in perspective with the optimal PV capacity forecast at 15.4 $GW_{peak}$ at the national level, coordinated investment between local and national actors is needed to prevent dispensable expenses. An uncoordinated design is expected to increase the total costs for residential energy systems from 12% to 83% and curtails 48% of local renewable electricity.

**Keywords:** energy communities; renewable energy hub; MILP; multi-objective optimization; Dantzig–Wolfe decomposition

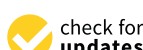



## 1. Introduction

In 2018, the European Parliament emphasized the role of energy communities in the energy transition [1]. The reasons include the penetration of renewable energies, the reduction of energy poverty and the enhancement of technological acceptance [2]. Energy communities aim at supplying energy needs with a high self-consumption of local energy sources. The reduction in the electricity grid reliance prevents costly grid reinforcements, therefore supporting a rapid electrification of heating and mobility services. In Switzerland, the electricity demand is expected to increase from 57 TWh/yr today to 95 TWh/yr in 2050, from which 33 TWh/yr is and will be supplied by hydropower [3,4]. Based on the cost-optimum scenario, the remaining electricity will be supplied by PV capacities (15.4 GW) and wind turbines (20 GW) [3]. The high investment into distributed capacities highlights the need for coordination between energy communities and grid utilities. The involvement of these actors in the decision-making dictates the energy flowing through the energy network, ultimately affecting the whole infrastructure.

The concept of an energy community is not strictly delimited, but it can be described as a local energy system encompassing distributed sustainable energy conversion units, both on the supply and demand sides [2]. The concept of the energy hub is usually used to model such systems. Multi-energy sources supply a multi-service demand with conversion units being optimally interconnected and operated. Extensive reviews have been carried out on this topic [5]. The scale considered varies from local energy hubs, such as a residential area to large-scale systems, including a whole country. Energy communities are usually deployed at the neighborhood scale since the proximity facilitates governance.

Due to its network structure, modeling an energy community at the district scale with a building resolution usually exceeds computational power [6]. Facing this problem, a popular method is to fix some degrees of freedom by making assumptions and scenarios based on expert knowledge (Table 1). As an example, half of the literature reviewed assumes energy demand profiles or predetermines the energy system configuration. The issue with such assumptions is the consideration of a fixed energy demand instead of energy services to be fulfilled. The change of approach is beneficial since it does not assume local investment decisions into energy capacities [7]. Therefore, it allows an optimal system design considering the interdependencies between subsystem components. An example within energy communities is the sharing of renewable energy capacities among buildings to maximize self-consumption. Another dimension is the coordination of the investments among subsystems to respect constraints at the district's boundary, such as grid constraints. Modeling subsystems as entities embedded in a larger system reveals the interdependency of the decision-making. It also entails an ethical aspect since the modeling should account for the interests of the actors concerned [7]. Therefore, assumptions and scenarios should be considered with care since they tend to oversimplify the view on the problem.

**Table 1.** Literature review on energy communities addressing simultaneously the optimal design and operation of energy systems: *Sub problem* and *Main problem* describe the resolution and problem boundary of the case studies. *Approach* shows the methodology used to handle problem complexity. *Interdependent decisions* highlights whether the authors considered decision-making interactions between buildings, and between the national and local scales. Finally, *Systemic constraints* refers to the boundary constraints consideration, such as grid constraints.

| Method | | | Analysis | | | |
|---|---|---|---|---|---|---|
| Sub Problem | Main Problem | Approach | National Scope | Interdependent Decisions | Systemic Constraints | Reference |
| Building | Building | Clustering | ✓ | ✗ | ✗ | [8] |
| Building | Building | Clustering | ✓ | ✗ | ✗ | [9] |
| Building | District | Pre-selection | ✗ | ✓ | ✗ | [10] |
| Building | District | Profiles | ✗ | ✗ | ✓ | [11] |
| Building | District | Pre-selection | ✗ | ✗ | ✗ | [12] |
| Building | District | Pre-selection | ✗ | ✗ | ✗ | [13] |
| Building | District | Dantzig-Wolfe | ✗ | ✓ | ✗ | [14] |
| District | District | Scenario | ✗ | ✗ | ✗ | [15] |
| Building | District | Scenario | ✗ | ✗ | ✗ | [16] |
| Building | District | Bi-level | ✗ | ✓ | ✓ | [17] |
| Building | District | Dantzig-Wolfe | ✗ | ✓ | ✗ | [18] |
| Building | District | Dantzig-Wolfe | ✗ | ✓ | ✗ | [19] |
| Building | District | Benders + Dantzig-Wolfe | ✗ | ✓ | ✗ | [20] |
| Building | District | Bi-level | ✗ | ✓ | ✓ | [21] |
| District | District | Rolling horizons | ✗ | ✗ | ✗ | [22] |
| Building | District | Clustering + Dantzig-Wolfe | ✓ | ✓ | ✓ | This paper |

Despite the extensive literature existing on the topic of energy communities, a holistic framework is usually not considered. Besides the simplification of the problem statement, another research gap is the lack of generality of the results. Most of the literature considers a single case study on a neighborhood. Some authors investigated the broad impact of local residential systems but the scope relied on single building energy systems [8,9]. Therefore, the potential of energy communities to support the energy transition at the national level is an evident knowledge gap in the literature. Based on these research gaps, the present study aims at answering the following research questions:

- Methodology for scaling-up local decisions to the national level:
    1. How to identify typical neighborhoods representing a whole country?
    2. How does the decision-making change with geographic and urban context?
- National systemic integration of local energy systems based on interface conditions:
    1. How does renewable electricity penetration change with electricity tariffs?
    2. What are the impacts of considering grid capacity for energy communities?

## 2. Methodology

The energy community is modeled as a renewable energy hub, being defined as a system optimally interconnecting multi-energy streams and conversion units [6]. Additionally, the energy hub is characterized by a high share of renewable energy and aims at maximizing self-consumption. The renewable energy hub is at the district scale within a low-voltage (**LV**) electricity grid served by a low-to-medium voltage (**LV/MV**) transformer (Figure 1). Service demands of each building, such as domestic hot water, domestic electricity and space heating, are supplied by conversion units and a gas and electricity utility. A mixed-integer linear programming (**MILP**) formulation optimizes the investment into conversion units and the operation of the energy system. The conversion units include thermal units (air–water heat pumps, gas boilers, and electrical heaters) and storage units (thermal tanks and lithium-ion batteries). Batteries are available both at the building and district scales. PV panels are the main source of renewable electricity. Their orientation on the roofs is a decision variable as described by Middelhauve et al. [6].

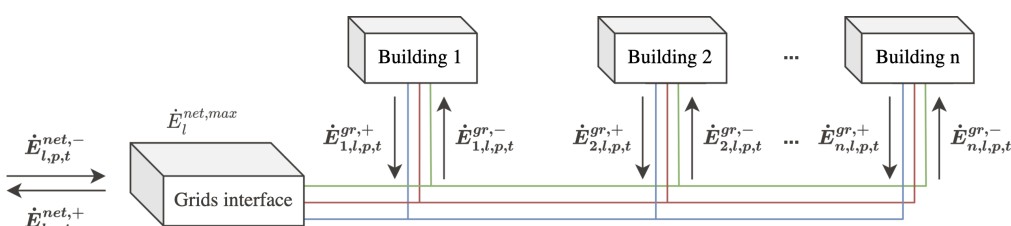

**Figure 1.** Energy community model with energy flows and network constraints.

### 2.1. Optimization Problem Formulation

The objective functions are described in Equations (1a)–(1c). In the equations, decision variables are highlighted with **bold** characters. The total costs (**TOTEX**) encompass operating costs (**OPEX**) and capital costs (**CAPEX**). The OPEX correspond to the annual energy costs and revenues. The electricity and gas retail tariffs are given by $c_l^+$, with $l$ either electricity or natural gas, and the feed-in tariff is $c_l^-$. The variables $E_{l,p,t}^{net,\pm}$ correspond to the energy exchange of an energy carrier $l$ over a period $p$ and a duration $t$ with the network outside the energy community. A positive symbol represents an import of energy, and a negative one, an export. The CAPEX (1c) consider investments and replacement costs of energy units. The costs are annualized over an $n$ years horizon with an interest rate $i$. The investment costs $C^{inv}$ are linearized with fixed ($i_u^{c1}$) and variable ($i_u^{c2}$) costs ((4a), (9a)). The CAPEX is dictated by two decision variables, the binary decision to install a unit or not ($y_u$) and the installed capacity ($f_u$). When a conversion unit has a lifetime $l_u$ lower than the project horizon $n$, the replacement cost is given by the number of replacements $R$

over the horizon $n$ ((4b), (9b)). Investment decisions are taken both at the district (4) and building levels (9). The size of each unit is lower- and upper-bounded by two reference capacities $F_u^{min}$ and $F_u^{max}$ (10), delimiting the region where the unit investment cost has a linear relationship with respect to its capacity. Multi-objective optimization is performed to evaluate the solution space at the interplay of two conflicting objectives: OPEX and CAPEX. One objective is upper-bounded by an $\epsilon$-constraint, while the second objective is minimized. Pareto fronts are generated by varying the $\epsilon$-constraints and by exchanging the objectives that are constrained and minimized:

$$\boldsymbol{TOTEX = OPEX + CAPEX} \tag{1a}$$

$$\boldsymbol{OPEX = \sum_{l \in L} \sum_{p \in P} \sum_{t \in T} c_l^+ \cdot E_{l,p,t}^{net,+} - c_l^- \cdot E_{l,p,t}^{net,-}} \tag{1b}$$

$$\boldsymbol{CAPEX = \frac{i(1+i)^n}{(1+i)^n - 1}(C^{inv} + C^{rep})} \tag{1c}$$

Energy and mass balances as well as heat cascade are the main constraints of the model. Equation (8a) shows the building energy balance between energy flows of the units $\dot{E}_{b,l,u,p,t}^{\pm}$, the domestic electricity demand $\dot{E}_{b,l,p,t}^{B,-}$ and the buildings import and export $\dot{E}_{b,l,p,t}^{gr,\pm}$. A second energy balance is applied at the district scale (3a), allowing synergies between buildings and between energy carriers. The energy balance is closed by energy exchanges with the network outside the community $E_{l,p,t}^{net,\pm}$. Technical constraints are considered to model conversion units and to account for infrastructure specifications. For example, Constraint (3b) is applied to restrict the power exchanged on the LV/MV transformer to a specified value $\dot{E}_{el}^{net,max}$. To reduce computational burdens, time series are clustered into typical and extreme operating periods. The typical day frequency is $d_p$, and the timesteps duration is $d_t$. The model considers five main sets: buildings $B$, layers of energy carriers $L$, typical periods $P$, timesteps $T$ of each typical period, and units $U$. More details on the formulation are given in the two following theses [6,8].

*2.2. Dantzig–Wolfe Decomposition*

Evaluating simultaneously investment and operation decisions at the building and district scales is computationally intensive due to the network structure of the problem. As described by Middelhauve et al. [23], the CPU time is around a few minutes for a problem with four buildings, whereas it reaches more than an hour with nine buildings. Therefore, the Dantzig–Wolfe decomposition is applied on the original MILP problem. The choice of this decomposition method is linked to the presence of the linking constraints (3), which are at the origin of the problem network structure. The algorithm has been described step by step by Middelhauve et al. [23]. Each building energy system represents a subsystem independent from other subsystems except for the resources balance at the district level (3a) and capacity constraints (3b), being linking constraints. The model is decomposed into two problems: a master problem (**MP**) and sub problems (**SPs**). The SPs represent the subsystems and contain building energy and mass balances, as well as investments in building-scale units. The MP considers linking constraints and represents the district energy system problem. Figure 2 describes the decomposition algorithm. First, the SPs are solved individually. The aim of the initiation is to obtain a set of system configurations representative of the SPs solution space. Each configuration accounts for an investment into conversion units $C_{i,b}^{inv/rep}$ and associated energy flows with the district grids $\dot{E}_{i,b,l,p,t}^{gr,\pm}$. A multi-objective optimization between CAPEX and OPEX is performed to obtain seven system configurations for each building. With this initial set of SPs configurations, the MP is initiated. The latter selects an optimal set of SPs configurations by a linear combination of the proposals. A weight $\lambda_{i,b}$ is attributed to each SPs design proposal. In the initiation and iteration loop, an LP relaxation is performed, and the MP is optimized with continuous $\lambda_{i,b}$. Once optimized, the MP calculates the dual values of the linking constraints that are inserted in the SPs objective function as Lagrangian multipliers. While the latter provide

SPs with information on the state of the district energy system, the SPs attempt to react to this signal with new design proposals. The SPs are formulated as reduced costs, meaning that a solution with a negative value has the potential to improve the MP objective. The iteration loop terminates when the SPs cannot find negative reduced costs or when the maximum number of iterations is reached. The solution obtained represents a lower bound due to the LP relaxation. Therefore, a last MP optimization is performed with binary weights $\lambda_{i,b}$ to ensure integrality of the SPs binary variables. This optimization provides an upper bound to the objective function. The resulting gap between the two bounds is typically below 0.1% [23]. This value is considered sufficiently low, and the algorithm is terminated.

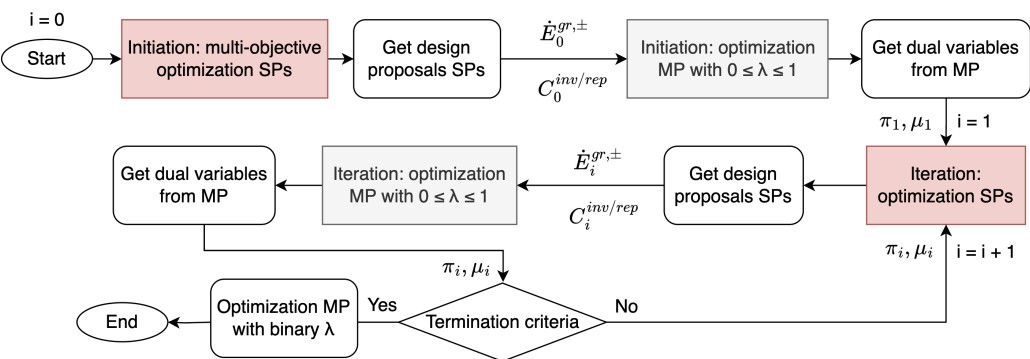

**Figure 2.** Dantzig–Wolfe decomposition algorithm with information flow between the MP and SPs.

### 2.2.1. Master Problem

The MP objective functions are the ones described in Equations (1a)–(1c). The main decision variable is the weight $\lambda_{i,b}$ attributed to each SPs design proposal $i \in I$. Convexity constraints (2) are applied, where Constraint (2a) represents the LP relaxation of the problem. A dual variable $\mu_b$ (5a) is associated with Constraint (2b) and represents the marginal cost of the building $b$ on the objective function. Energy balances and capacity constraints represent the linking constraints (3) and are associated with the dual variable $\pi_{l,p,t}$. The latter corresponds to the marginal cost profile of resources in the district and is inserted as an energy tariff in the SPs operating cost function (7). Investment costs (4) account for units at the building ($u \in U$) and district ($u \in U^*$) scales:

$$0 \le \lambda_{i,b} \le 1 \quad \forall i, b \in \mathrm{I}, \mathrm{B} \tag{2a}$$

$$\sum_{i \in \mathrm{I}} \lambda_{i,b} = 1 \quad \forall b \in \mathrm{B} \quad \frown [\mu_b] \tag{2b}$$

$$\sum_{i \in \mathrm{I}} \sum_{b \in \mathrm{B}} \lambda_{i,b} \cdot \left( \dot{E}^{gr,+}_{i,b,l,p,t} - \dot{E}^{gr,-}_{i,b,l,p,t} \right) \cdot d_p \cdot d_t = E^{net,+}_{l,p,t} - E^{net,-}_{l,p,t} \quad \frown [\pi_{l,p,t}] \tag{3a}$$

$$\dot{E}^{net,\pm}_{l,p,t} \le \dot{E}^{net,max}_l \qquad \forall l, p, t \in \mathrm{L}, \mathrm{P}, \mathrm{T} \tag{3b}$$

$$C^{inv} = \sum_{i \in \mathrm{I}} \sum_{b \in \mathrm{B}} \lambda_{i,b} \cdot C^{inv}_{i,b} + \sum_{u \in U^*} b_u \cdot (i^{c1}_u \cdot y_u + i^{c2}_u \cdot f_u) \tag{4a}$$

$$C^{rep} = \sum_{i \in \mathrm{I}} \sum_{b \in \mathrm{B}} \lambda_{i,b} \cdot C^{rep}_{i,b} + \sum_{u \in U^*} \sum_{r \in R} \frac{1}{(1+i)^{r \cdot l_u}} \cdot (i^{c1}_u \cdot y_u + i^{c2}_u \cdot f_u) \tag{4b}$$

$$[\mu_b] = \frac{\Delta \mathrm{obj}}{\Delta \left( \sum_{i \in \mathrm{I}} \lambda_{i,b} \right)} \quad \forall b \in \mathrm{B} \tag{5a}$$

$$[\pi_{l,p,t}] = \frac{\Delta \mathrm{obj}}{\Delta \left( E^{net,+}_{l,p,t} - E^{net,-}_{l,p,t} \right)} \quad \forall l, p, t \in \mathrm{L}, \mathrm{P}, \mathrm{T} \tag{5b}$$

### 2.2.2. Sub-Problem

The objective function of the the SPs (6) is the reduced cost of each building energy system. The main decision variables are the sizing of the energy units ($y_{b,u}$ and $f_{b,u}$) and energy flows with the grid $\dot{E}^{gr,\pm}_{b,l,p,t}$. Similar to Constraint (3b), the LV lines capacity can be considered with Constraint (8b), where $\dot{E}^{gr,max}_{b,l}$ corresponds to the maximum connection power of a building $b$. Equation (11a) further details the space heating demand. Buildings are represented with a 1R1C thermal model, where $U_b$ is the heat transfer coefficient of the building and $C_b$ is its heat capacity. The indoor temperature $T^{int}_{b,p,t}$ is a decision variable, allowing for building pre-heating. Space heating requirements are divided into $K$ temperature intervals $\dot{R}^{\pm}_{k,b,p,t}$ and are supplied by hot streams from energy conversion units following heat cascades ((11b), (11c)):

$$Min \quad C^{op}_b + \frac{i(1+i)}{(1+i)^n - 1}(C^{inv}_b + C^{rep}_b) - \mu_b \tag{6}$$

$$C^{op}_b = \sum_{l \in L} \sum_{p \in P} \sum_{t \in T} \left( \pi_{l,p,t} \cdot \dot{E}^{gr,+}_{b,l,p,t} - \pi_{l,p,t} \cdot \dot{E}^{gr,-}_{b,l,p,t} \right) \cdot d_t \cdot d_p \quad \forall b \in B \tag{7}$$

$$\dot{E}^{gr,+}_{b,l,p,t} + \sum_{u \in U} \dot{E}^{+}_{b,l,u,p,t} = \dot{E}^{gr,-}_{b,l,p,t} + \sum_{u \in U} \dot{E}^{-}_{b,l,u,p,t} + \dot{E}^{B,-}_{b,l,p,t} \tag{8a}$$

$$\dot{E}^{gr,\pm}_{b,l,p,t} \leq \dot{E}^{gr,max}_{b,l} \qquad \forall b, l, p, t \in B, L, P, T \tag{8b}$$

$$C^{inv}_b = \sum_{u \in U} b_u \cdot (i^{c1}_u \cdot y_{b,u} + i^{c2}_u \cdot f_{b,u}) \qquad \forall b \in B \tag{9a}$$

$$C^{rep}_b = \sum_{u \in U} \sum_{r \in R} \frac{1}{(1+i)^{r \cdot l_u}} \cdot (i^{c1}_u \cdot y_{b,u} + i^{c2}_u \cdot f_{b,u}) \quad \forall b \in B \tag{9b}$$

$$y_{b,u} \cdot F^{min}_u \leq f_{b,u} \leq y_{b,u} \cdot F^{max}_u \qquad \forall b, u \in B, U \tag{10a}$$

$$f_{b,u,p,t} \leq f_{b,u} \qquad \forall b, u, p, t \in B, U, P, T \tag{10b}$$

$$\dot{Q}^{SH}_{b,p,t} = \dot{Q}^{gain}_{b,p,t} - U_b \cdot A^{era}_b \cdot (T^{int}_{b,p,t} - T^{ext}_{p,t}) - C_b \cdot A^{era}_b \cdot (T^{int}_{b,p,t+1} - T^{int}_{b,p,t}) \tag{11a}$$

$$\forall b, p, t \in B, P, T$$

$$\dot{R}_{k,b,p,t} - \dot{R}_{k+1,b,p,t} = \sum_{u_h \in S_h} \dot{Q}^{-}_{u_h,k,b,p,t} - \sum_{u_c \in S_c} \dot{Q}^{+}_{u_c,k,b,p,t} \tag{11b}$$

$$\dot{R}_{1,b,p,t} = \dot{R}_{n_k+1,b,p,t} = 0 \qquad \forall k, b, p, t \in K, B, P, T \tag{11c}$$

### 2.3. Limitations of the Model

The aim of the model is to design an energy system by optimally selecting the capacity of energy conversion and storage units. The optimization takes peak power into account when sizing energy networks. However, the dynamics of the power system, such as frequency and voltage stability, are not considered. As a result, the design of the energy infrastructure is based on an estimate of the peak power that would occur during its operation. A second limitation of the model is its deterministic nature, which can be offset by sensitivity analysis.

### 2.4. Key Performance Indicators

Key performance indicators are used to quantify the solutions performance. The self-consumption (**SC**) is the share of onsite electricity generation potential $E^{pot}$ being consumed within the district (12a). The self-sufficiency (**SS**) corresponds to the share of the electricity demand being supplied by onsite generated electricity (12b). PV curtailment is the share of onsite electricity generation potential being neither self-consumed, nor sold to

the grid (12c). Finally, the PV penetration (**PVP**) is the proportion of the electricity demand that could be supplied by the onsite generated electricity with a SC of 100% (12d). The global warming potential (**GWP**) accounts for both the energy system construction and operation emissions (12e) as described in [6]. Emissions related to the installation of energy conversion units ($i_u^{g1}$ and $i_u^{g2}$) are taken from the Ecoinvent 3.6 database with the method IPCC 2013.

$$\text{SC} = (E^{pot} - E^{curt} - E_{el}^{net,-})/E^{pot} \tag{12a}$$

$$\text{SS} = (E^{pot} - E^{curt} - E_{el}^{net,-})/(E^{pot} - E^{curt} - E_{el}^{net,-} + E_{el}^{net,+}) \tag{12b}$$

$$\text{PVC} = E^{curt}/E^{pot} \tag{12c}$$

$$\text{PVP} = E^{pot}/(E^{pot} - E_{el}^{net,-} + E_{el}^{net,+}) \tag{12d}$$

$$\text{GWP} = \sum_{l \in L}\left(g_l^+ \cdot E_l^{net,+} - g_l^- \cdot E_l^{net,-}\right) + \sum_{u \in U}\frac{1}{l_u} \cdot \left(i_u^{g1} \cdot y_u + i_u^{g2} \cdot f_u\right) \tag{12e}$$

### 2.5. Typical Districts Identification

Gupta et al. estimated that Switzerland hosts 17'844 MV/LV transformers [24]. To handle this problem complexity, a k-medoids clustering algorithm is applied to find the most representative districts of the country. The case study is implemented through a geographic information system to adequately describe the energy demands and sources. Clustering features consider real-estate characteristics (heating surface, roof area, service demands, building category, and construction year) and geographic ones (annual solar irradiation, average temperature, electricity and gas grids density). Typical Swiss weather profiles have been assessed for each district by Stadler et al. [8]. A principal component analysis is applied to reduce the dataset dimensionality [25]. The purpose is to restrain the computational time while keeping the information variability from the dataset.

The k-medoids algorithm is run over 50 iterations to estimate the optimal number of clusters $K_{opt}$. The latter corresponds to the minimum number of clusters required to represent the initial dataset as faithfully as possible. Three indicators are used to assess the clustering quality: elbow, silhouette and Calinski–Harabasz (CH). The silhouette and CH indexes are two metrics of the clusters cohesion [26]. Therefore, the aim is to maximize their score. In contrast, the elbow method measures the distortion of the clusters and should be minimized. Figure 3 presents the mean score of the indicators. The trade-off between information loss and complexity reduction is clear for the CH and elbow index. Below 10 clusters, the marginal performance improvement of adding a new cluster is high. The value of $K_{opt}$ is, respectively, 6 and 7 for the elbow and CH index. Regarding the silhouette index, $K_{opt}$ reaches 4 clusters but the metric does not capture well the trade-off between information loss and complexity. While Figure 3 shows the mean score of the metrics over all iterations, Figure 4 represents the distribution of $K_{opt}$. The silhouette and elbow score are robust. However, the result of the CH metric is distributed over several $K_{opt}$, ranging from 12 to 19 clusters. Since $K_{opt}$ is measured at the maximum of the CH score and since this maximum is located on a plateau, it makes $K_{opt}$ difficult to locate. Therefore, the mean score interpretation is preferred. Based on this analysis, the optimal number of clusters is between 6 and 10. The exact value of $K_{opt}$ depends on the level of detail needed for the case studies to accurately represent specific regions. This clustering algorithm allows considering versatile district typologies within a restricted set of case studies. In particular, it allows the contextualization and extrapolation of local decisions to the national scale.

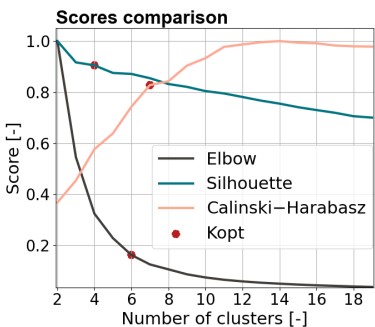

**Figure 3.** Mean scores obtained for the three metrics over 50 iterations.

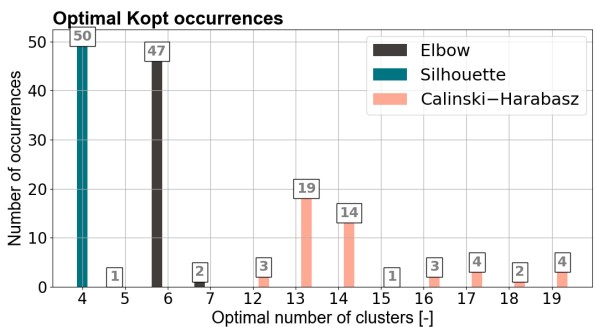

**Figure 4.** Occurrences of the optimal number of clusters.

### 2.6. Case Study

Within this case study, 6 clusters are identified to represent the whole Swiss building stock (Figure 5). The Urban cluster is the largest one with a mix of single and multi-family dwellings, plus commercial centers. The Sub-urban one is represented by a compact village of 2-floor buildings. The Countryside and Mountain areas are both dominated by single-family detached houses but with different densities. Finally, the Forest cluster is dominated by large farms converted into residential houses. The sixth cluster is not considered in the case study since it is characterized by forests, mountains and glaciers without any buildings. For extrapolation to the national level, the representative roof area of each typical district is used. Since the present study aims at analyzing the impact of energy communities, it is assumed that each district in Switzerland can endorse the status of an energy community. Figure 6 shows the convergence curves of each typical district for a TOTEX minimization. The urban cluster requires the largest number of iterations to converge. One possible explanation is linked to the heterogeneity of the buildings and their large number.

Most data are open source and provided by the Swiss government. The building characteristics, such as the height, heated areas or types of construction, come from the cantonal and federal Official Buildings Registry [27]. Energy standards such as the envelope heat transfer, building heat capacity, and domestic electricity demand, as well as the internal and external heat gains are calculated based on Swiss standard norms [28]. These data are used to build the thermal model of the buildings [29]. The outdoor temperature and solar irradiation come from Meteonorm [30]. These time series are clustered into ten typical periods of 24 h and two extreme periods of one hour using k-medoids clustering. The project horizon is 20 years, and the interest rate 2%. Electricity and gas retail tariffs are, respectively, 0.25 CHF/kWh and 0.14 CHF/kWh and the feed-in tariff is 0.10 CHF/kWh. These values are based on average energy tariffs in Switzerland for the years 2022–2023 [31]. The carbon content of electricity and natural gas are respectively set to 0.13 and 0.23 kg $_{CO_2}$/kWh. The fixed investment cost for PV panels is CHF 6556, and the variable cost is 1300 CHF/kW$_{peak}$. Regarding batteries, the fixed cost is CHF 825, and the variable cost 1290 CHF/kWh. More details on the parameters settings are provided by Middelhauve et al. [6].

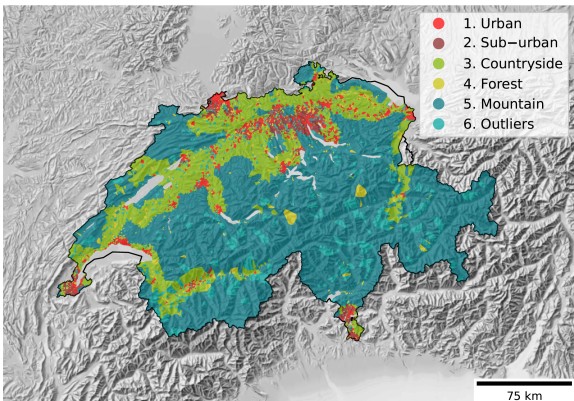

**Figure 5.** Typical districts distribution in Switzerland. The clusters differentiate the morphological and meteorological characteristics of the Swiss building stock.

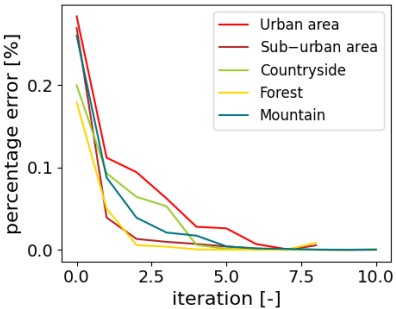

**Figure 6.** Convergence curves: percentage error between lower and upper bounds with a maximum of 10 iterations.

## 3. Results and Discussion

The discussion follows two axes. First, the decision-making trends within energy communities are analyzed and contextualized with their geographic and urban characteristics. Then, the potential of energy communities to supply renewable electricity to the national infrastructure is analyzed. The analysis considers grid constraints and confronts the cost and energy efficiency impacts of a coordinated or uncoordinated investment strategy between local investments and national energy needs.

### 3.1. Decision-Making Trends within Energy Communities

A multi-objective optimization between the CAPEX and OPEX is performed to obtain the solution trends for each district. Figure 7 shows the progressive substitution of the principal energy source, natural gas, by renewable energy as investments are increased. Gas boilers deployment corresponds to the solution with lowest investment and highest operational cost. Then, it is substituted by investments into heat pumps and solar panels. The marginal PVP improvement decreases with investment since the best roof orientations are activated first. The typical districts split into three decision-making trends. This behavior is mainly explained by the ratio $\eta_s$ between roof surfaces and heated surfaces. This ratio can be considered as an approximation of the PVP. With its large farms, the Forest cluster (4) has the largest solar potential. However, it has as well the highest heated surface among the typical districts resulting in the lowest $\eta_s$ ratio ($\eta_s = 0.57$) and therefore the lowest PVP. The Urban (1) and Sub-urban (2) districts have similar building compactness ($\eta_s = 0.80$ and $0.86$), even though they contain different building density and usage purposes. Finally, since clusters 3 and 5 mainly contain single-family dwellings, they have the largest roof area with respect to their energy demand ($\eta_s = 1.26$ and $1.42$). Since a large $\eta_s$ ratio usually represents small buildings, it means as well a low economy of scale.

As there is a fixed investment cost to install energy units, the investment to decarbonize the energy system increases together with $\eta_s$.

Besides decarbonization, local investment decisions have as well an impact at the districts boundary. Figure 8 presents the annual electricity flows at the LV/MV transformers. Electricity imports remain more or less constant with the investment since the installation of PV panels compensates for the electricity consumption of heat pumps. In District 4, heat pumps are deployed before PV panels, resulting in an increase in electricity imports in the low-investment region. Electricity exports follow the same trend as the PVP, with the difference of exhibiting a constant slope of 9.4 kWh/CHF, a value to be put in perspective with the feed-in tariff promoting a sale of 10 kWh/CHF. It demonstrates the sensitivity of investments and energy flows to energy tariffs. Besides maximizing self-consumption, energy communities are able to move from passive consumers to renewable electricity suppliers. And this shift in role is predetermined based on the price signals. A question still remains: do the grid operators provide the right incentives to perform the energy transition?

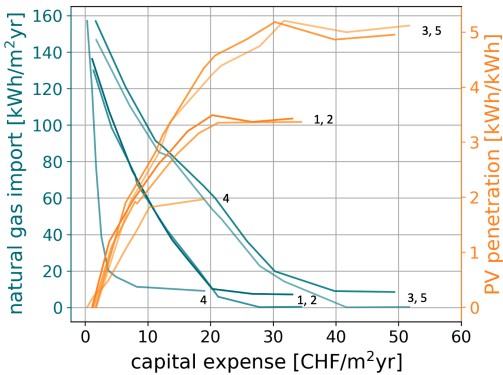

**Figure 7.** Natural gas imports and renewable electricity generated onsite for each typical district.

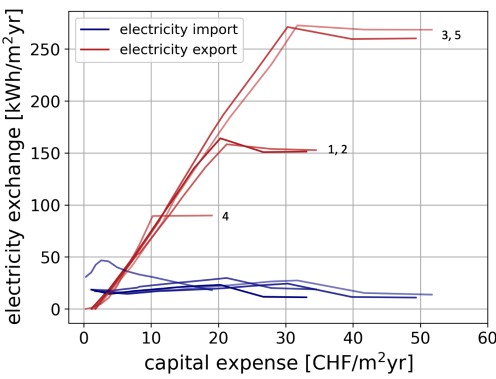

**Figure 8.** Electricity flows at the districts LV/MV transformer, normalized by heated surface.

### 3.2. National-Scale Impacts of Energy Communities

As seen in Figure 7, the marginal cost of PV panels is increasing with the investment allocated. The last economically feasible point is obtained once the investments into PV panels break even with the revenues over their lifetime [6]. Since the revenues vary with the electricity tariffs, the investments induced by the tariffs can be calculated assuming that local stakeholders would invest until they reach the last economically feasible point. Figures 9 and 10 present these induced investments in the form of annual renewable electricity generated by energy communities in Switzerland for a range of feed-in and retail tariffs. Figure 9 is generated assuming that the investments were optimized for each building individually while Figure 10 represents the case of energy communities. Below a certain energy tariff, PV investments are not profitable due to the affordable electricity cost from the grid. The investment threshold is delimited by the lower black line. On the other side,

the upper investment limit maps the region where the PV installed capacity reaches its maximum of 67.2 GW, representing an annual electricity generation of 80 TWh/yr. The extrapolation to the national scale assumes that 70% of the roofs are suitable for PV installations [32]. The PV potential calculated in this study is 14% higher than the one estimated by Swissolar (70 TWh/yr [32]). Since the PVP varies within and throughout typical districts, there exists a large spectrum of investments. Districts with high solar potential are firstly selected. Then investments with lower profitability are allocated as the price signals sent by the grid operators become more attractive. Present energy tariffs incite to invest into the maximum PV panels capacity, reaching a potential of 80 TWh/yr. However, Schnidrig et al. estimated that the cost-optimal PV deployment in Switzerland would lead to a 20 TWh/yr electricity generation from PV panels [3]. This optimum considers the possibility of energy storage in hydro dams, batteries, and biogas or hydrogen reservoirs. In addition, it accounts for grid availability, power losses and peak power. Based on this value, the price incentives should fall within the red–orange region. In conclusion, there is a discordance between the price signals sent by grid operators and the long-term needs of the infrastructure. This situation could result into costly grid reinforcements or local PV generation curtailment. In both cases, the solution is socially unfair since the former induces costs to end users and the latter might render some investments unprofitable. In the remaining of the analysis, the impact of curtailment is analyzed in terms of energy efficiency and costs.

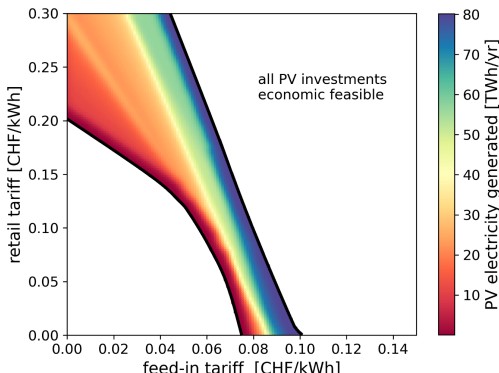

**Figure 9.** Optimal yearly renewable electricity generation based on building scale decisions.

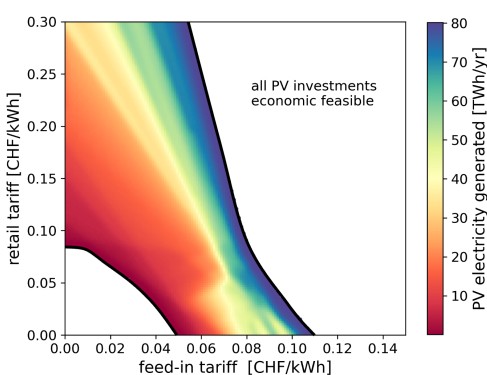

**Figure 10.** Optimal yearly renewable electricity generation based on energy communities.

To support the analysis, two scenarios are considered. In the first one (uncoordinated), an investment decision into PV panels and heat pumps is taken today. Then, peak shaving is applied on the energy system. The investment into storage units and their operation are optimized with fixed sizes of PV and heat pump units. In the second scenario (coordinated), all investment and operation decisions are taken considering peak shaving. Therefore, PV and heat pump capacities vary as the system is being constrained. The aim of these two scenarios is to assess the impact of peak shaving on the decision-making and to measure the importance of coordinating investments. Figure 11 shows the load duration curve of

electricity flows between energy communities and the national grid. The present electricity tariffs promote a peak export power of 46.8 GW, exceeding by a factor of three the existing capacity of the Swiss MV grid, being 15.8 GW [3]. The optimal PV capacity is represented in Figure 11 by the blue area and corresponds to a peak PV power of $15.4 \pm 2$ GW with a self-consumption of 5.8 GW at the time of the peak export. The curtailed system reduces by two thirds the maximum export power. The uncoordinated scenario exports more electricity and has a flat profile. Because of its large PV capacity, the energy system simply curtails the exceeding power. This outcome is beneficial for the grid operators since the export profile shows fewer variations. However, from the perspective of households, the PV investment is oversized since the optimal export profile would have been 29% lower (plain red line).

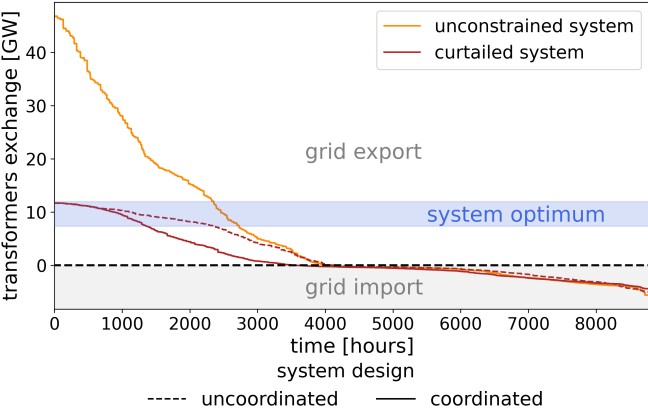

**Figure 11.** Load duration curve of electricity imports and exports for energy communities in Switzerland. The unconstrained solution is constrained to reduce the maximum power peaks by a factor of three. Two design scenarios are considered, a coordinated one accounting for peak shaving in the investment decision and an uncoordinated one, with peak shaving being imposed after the investment decision.

For each scenario, Figure 12 further details the energy efficiency (A), the energy units installed (B) and the associated energy flows and GWP (C). Each plot represents the impacts on energy communities as the peak export power is increasingly constrained, from 46.8 GW down to 5.1 GW as shown on Figure 11. In the uncoordinated scenario, an investment into 35 GWh$_{th}$ of thermal storage is allocated to increase self-consumption and self-sufficiency. In addition, 0.12 GWh$_{el}$ of batteries is installed. Even though the energy storage strategy increases the energy performance, it is not sufficient to absorb the large amount of electricity generated onsite. Therefore, most of the peak shaving is performed by PV curtailment, and 48% of the renewable electricity is simply not used. In contrast, under the consideration of peak shaving, the coordinated scenario invests 37% less into PV panels. It reduces the self-sufficiency by 20%, but it mitigates curtailment with a PVC of 9% only. In addition, the reduction in PV capacity and grid constraints decrease the electrification of space heating by 9%. Consequently, the use of gas boilers as ancillary units during peak demand hours increases the GWP by 17%. It is worth mentioning that alternative solutions, such as district heating, synthetic gas storage or the use of biomass, could provide the offloading necessary to decarbonize hard-to-abate emissions. Although the methodology makes it possible to incorporate new technological breakthroughs, these are not yet included in this analysis.

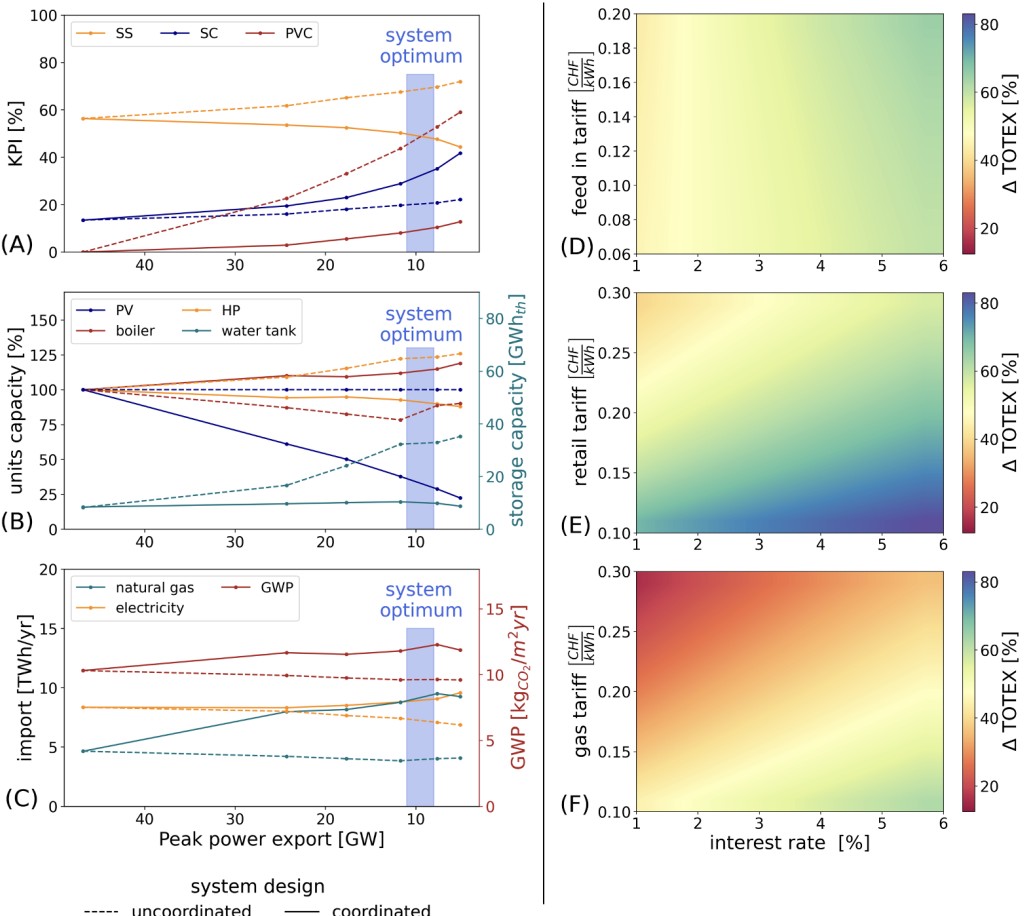

**Figure 12.** (**A**) Energy performance, (**B**) energy units capacity and (**C**) energy imports and GWP with respect to peak shaving. All values are given after extrapolation to the whole country. The GWP is normalized per heated surfaces and considers the embodied emissions and emissions of purchasing energy carriers. (**D**–**F**) present the total cost differences between the two system designs at the peak shaving optimum for a range of energy tariffs and interest rates.

The two system designs differ in terms of unit capacity and energy exchanges with the networks. They, therefore, lead to a difference in financial performance that depends on the economic context. Figure 12D–F present this financial gap for a range of interest rates and energy tariffs. The oversized PV capacity of the uncoordinated design and its PV curtailment induce a total cost increase from 12% to 83% compared to the coordinated design. Increasing energy tariffs has the tendency to decrease the financial gap due to the low amount of energy imports in the first scenario. In particular, high natural gas prices impact the economic performance of the coordinated scenario due to its use of gas boilers during peak demand hours. In contrast to energy tariffs, the interest rate sharpens the economic gap due to the large investments in the uncoordinated scenario.

## 4. Conclusions

The objective of this paper is to highlight the decision-making trends within energy communities and their integration in the national energy infrastructure. The communities are modeled as renewable energy hubs, considering operation and investments into energy conversion units. Five typical districts are considered, and the energy system solutions are extrapolated to the national scale. Multi-objective optimization and grid constraints are applied to fulfill the national needs for local renewable electricity. The main outcomes of the study are as follows:

- Investment trends are similar among the typical districts. However, their magnitude and solar potential differ based on the location and morphology of the buildings.
- The methodology provides a good estimation of the solar potential in Switzerland with a limited set of typical districts. The estimation is 14% above the findings of previous detailed studies [32].
- Investment and operation decisions in energy communities are highly sensitive to electricity tariffs. Present price signals promote an excessive PV deployment into the energy system, with an installed capacity that could considerably exceed by a factor of three the forecast cost optimum of 15.4 GW [3].
- Uncoordinated investments with respect to grid constraints could generate curtailment up to 48% and increase total costs from 12% to 83%. In contrast, a coordinated planning where energy communities adapt their equipment to the specifications of the infrastructure only curtails the PV generation potential by 9%.

The presented results contribute to a better understanding of the decision-making interdependency between local actors and national energy systems. A holistic approach encompassing various stakeholders within a single optimization framework favors a coordinated energy transition and increases acceptance for the decision makers. Grid operators and national institutions should adopt a consensus on the appropriate price signals to send to local stakeholders to incentivize renewables deployment while preventing flawed returns on investment. The extension of the work includes a better definition of the national infrastructure. To this extent, bi-level and nested decomposition methods have a high potential to link the various levels of decision-making.

**Author Contributions:** Conceptualization, C.T.; Data curation, C.T., J.R.H.L. and D.L.; Formal analysis, C.T.; Funding acquisition, F.M.; Investigation, C.T.; Methodology, C.T., J.R.H.L., D.L. and F.M.; Project administration, F.M.; Resources, F.M.; Software, C.T., J.R.H.L. and D.L.; Supervision, F.M.; Validation, J.R.H.L. and D.L.; Visualization, C.T. and J.R.H.L.; Writing—original draft, C.T.; Writing—review and editing, J.R.H.L. and D.L. All authors have read and agreed to the published version of the manuscript.

**Funding:** The research published in this report was carried out with the support of the Swiss Federal Office of Energy SFOE as part of the SWEET consortium SWICE. The authors bear sole responsibility for the conclusions and the results of the presented publication.

**Data Availability Statement:** The data in this study was generated using the open source optimization tool REHO. The reader can download the Git repository and generate similar data according to the documentation available here: https://reho.readthedocs.io/en/main/ (accessed on 5 February 2024).

**Conflicts of Interest:** The authors declare no conflicts of interest. The funders had no role in the design of the study; in the collection, analyses, or interpretation of data; in the writing of the manuscript; or in the decision to publish the results.

## Abbreviations

The following abbreviations are used in this manuscript:

| | | | |
|---|---|---|---|
| LV/MV | Low voltage/medium voltage | GWP | Global warming potential |
| CAPEX | Capital cost | PVP | Photovoltaic penetration |
| OPEX | Operating cost | SC | Self-consumption |
| TOTEX | Total cost | SS | Self-sufficiency |
| MP/SPs | Master/sub problems | PVC | Photovoltaic curtailment |

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
