# Peer review of "From Local Energy Communities towards National Energy System: A Grid-Aware Techno-Economic Analysis"

_energies, doi:10.3390/en17040910_

Round 1
Reviewer 1 Report
Comments and Suggestions for Authors
The paper addresses the integration of energy communities at the grid level with a bottom-up approach. District energy systems were modeled using a mixed integer linear programming approach. The Dantzig-Wolfe decomposition was applied to decrease the computational effort. The methodology considers a framework comprising PV energy facilities.
The text is well organized, clearly written, and with suitable use of the English language, regarding a topic of current interest within the scope of the journal. Moreover, a comprehensive literature review on energy communities was presented where the authors´ contributions were classified according to the scope, the methodology, the integration scale, and boundary constraints.
I pose some suggestions for the authors´ consideration on the final version of the text.
Formal aspects:
[1] Reduce keywords to five terms and consider using ‘Dantzig-Wolfe decomposition’ instead of simply ‘decomposition’ in lines 20 and 21.
[2] Change the title of item 2.1 to “Dantzig-Wolfe decomposition” in line 120.
[3] Reference 4 does not seem to be correctly presented in line 363.
Comments on content:
[1] The energy community is modeled as a renewable energy hub, being defined as a system that interconnects multi-energy lines and photovoltaic conversion units. The hub structure despises completely the possibility of Energy Storage Systems (ESS) that are each time more technically available and economically feasible. For instance, the Moss Landing (California, US) is presently the World’s biggest battery storage asset with 3GWh storage capacity and rated power at 750 MW. Systems like that can store renewable intermittent energy such as solar and wind. This feature changes strongly the assumptions made in this study yielding completely different conclusions. As far as I am concerned, the authors should, at least, cite the possibility of energy storage and highlight the fact that this resource was neglected.
[2] The methodology assumptions neglect the influence of environmental, social, and political aspects such as UN sustainable development goals, political instability on gas importation, and new technological breakthroughs.
[3] Despite the good quality of the investigation, the proposed approach is based on financial performance parameters assuming a certain scenario. Scenarios are always subject to uncertainties not covered by the investigation.
The paper treats the integration of energy communities as a linear programming problem, aiming to minimize financial costs (OPEX and CAPEX) and computation effort. The mathematical approach is clear and correct, however, I think that the grid constraints were too simplified, disregarding transmission losses, operation stability, and grid availability. Furthermore, social, and environmental aspects were neglected.
Comments on the Quality of English LanguageComments on English:
[1] Rephrase “2. What is (are) the impacts …” in line 76.
[2] I suggest that the authors adopted either American English (Am.E.) or British English (Br.E.) for grammar and spelling, e.g., the authors use “optimization” (Am.E.) instead of “optimsation” (Br.E.), but “modelled” (Br.E.) is used instead of “modeled” (Am.E.), contextualisation (Br.E.) instead of contextualization (Am.E.), analysed (Br.E.) vs analyzed (Am.E.), contextualised (Br.E.) vs contextualized (Am.E.), and so forth. I suggest a complete revision on Am.E versus Br.E. to adopt an only standard.
[3] Rephrase “Beside(s) decarbonization, local investment decisions …” in line 232.
Reviewer 2 Report
Comments and Suggestions for Authors
This paper investigates the decisions-making trends within energy communities and their integration in the national energy infrastructure. It’s an interesting topic and the authors make a good contribution to the field. However, there are some weaknesses in the paper, and the authors are suggested to solve these weaknesses to improve the quality of the paper.
1). The novelty of this paper is not very clear. It is yet not clear to me what main contributions of this paper compared to the large body of literature on the topic are. The authors are encouraged to clearly point out the contributions of this work points by points.
2). Please provide a more comprehensive literature review about the integration of renewable power generation in microgrids and energy communities. For example, the following important studies (doi: 10.1109/TSG.2022.3193030, doi: 10.1109/TSG.2022.3153230) that are associated with renewable power generation in local energy communities or microgrids are not mentioned in the Introduction.
3) Why are the energy balance constraints not considered in this paper? Please provide more explanations.
4). It is not clear what are the master problem (MP) and several sub problems (SPs). It is suggested to provide the detailed models of MP and SPs.
5) Besides, it is suggested to provide the iteration steps of the employed algorithms.
6). The authors are suggested to provide the convergence curve of the employed algorithm in the case study.
Reviewer 3 Report
Comments and Suggestions for Authors
Dear authors,
Thank you very much for your interesting contribution. The article proposes a methodology for the technical-economic analysis of the impact of the massive integration of energy communities into the national energy infrastructure of Switzerland. The proposed methodology can be applied in other national contexts similar to the one described.
The article is well-developed. In the introduction, the main objectives and the context of the research are described. As a suggestion, the authors might consider including some graphical diagrams to help understand the formulation described by equations (1a) and (1b).
It could also be interesting to detail the methodological limits of the proposal or under what conditions the methodology does not perform adequately, so that the weak points of the proposal are made explicit.
Best Regards
Round 2
Reviewer 1 Report
Comments and Suggestions for Authors
All comments I have posed were properly addressed by the authors, thus I recommend the paper acceptance in its present form.
Author Response
Dear reviewer,
We thank you for the previous comments you brought to our paper.
Please note that on the basis of the comment of another reviewer, we have brought additional details on the problem mathematical formulation (pages 5-6).
Sincerely,
Cédric Terrier, Joseph Loustau, Dorsan Lepour, and François Maréchal.
Reviewer 2 Report
Comments and Suggestions for Authors
Thank you for the response from the authors. However, the majority of the concerns raised by the reviewer have not been addressed. I will now outline these concerns:
1) This study does not take into account power flow network constraints for both energy communities and the national energy system. This oversight could result in an unrealistic solution. Therefore, the solution and conclusions presented, without considering practical network constraints, lack conviction.
2) The author states that the Dantzig-Wolfe algorithm is not a contribution. However, its comparison is included in Table 1, which is confusing.
3) Is it not possible to solve the model directly using Cplex or Gurobi? Why was the Dantzig-Wolfe algorithm chosen instead of the Bender decomposition algorithm? Many studies suggest that the Bender decomposition algorithm outperforms the Dantzig-Wolfe algorithm. It is recommended to provide comparisons between the two.
4) Although the paper claims to aim at optimally selecting the capacity of energy conversion and storage units, no relevant constraints are provided to support this claim.
5) Furthermore, the model only includes an objective function and lacks constraints. This renders the model meaningless from a mathematical optimization perspective.
6) Firstly, the paper fails to provide the master problem (MP) and several subproblems (SPs). Secondly, there is no description of the iteration steps used in the employed algorithms. Additionally, the convergence curve of the employed algorithm is not provided. These omissions make the model highly unclear, and the results are not convincingly presented. It is suggested to provide the code used in the paper.
